# Physical activity, acute severity and long-term consequences of COVID-19: an 18-month follow-up survey based on a Swedish national cohort

Annie Palstam [ID],[1,2,3] Johanna Seljelid,[1] Hanna Charlotte Persson,[1,4] Katharina S Sunnerhagen [ID] [1,2]

[1]Department of Clinical Neuroscience, Institute of Neuroscience and Physiology, Sahlgrenska Academy, University of Gothenburg, Goteborg, Sweden
[2]Department of Neuroscience, Sahlgrenska University Hospital, Goteborg, Sweden
[3]School of Health and Welfare, Dalarna University, Falun, Sweden
[4]Department of Occupational Therapy and Physical Therapy, Sahlgrenska University Hospital, Goteborg, Sweden

**Correspondence to**
Dr Annie Palstam;
annie.palstam@gu.se

## ABSTRACT

**Objective** To investigate how changes in levels of physical activity (PA) in regard to acute disease severity relate to perceived difficulties in performing daily life activities 18 months after COVID-19 infection.

**Design** An observational study with an 18-month follow-up survey based on registry data from a national cohort.

**Participants** 5464 responders to the 18-month follow-up survey of a Swedish national cohort of 11 955 individuals on sick leave due to COVID-19 during the first wave of the pandemic.

**Outcomes** The follow-up survey included questions on daily life activities, as well as present and retrospective level of PA. Changes in PA level from before COVID-19 to follow-up were assessed by the Saltin-Grimby PA Level Scale and analysed by the Wilcoxon signed-rank test. Comparisons of groups were analysed by the Student's t-test, Mann-Whitney U test and $\chi^2$. Multiple binary logistic regression was performed to assess the association of changes in PA with perceived difficulties in performing daily life activities.

**Results** Among the 5464 responders (45% of national cohort), the PA level decreased. Hospitalised individuals had a lower PA level both prior to COVID-19 (p=0.035) and at the 18-month follow-up (p=0.008) compared with non-hospitalised responders. However, the level of PA decreased in both groups. A decrease in PA level increased the odds (OR 5.58, 95% CI 4.90 to 6.34) of having difficulties performing daily life activities.

**Conclusions** PA levels were reduced 18 months after COVID-19 infection. A decrease in PA over that time was associated with perceived difficulties performing daily life activities 18 months after COVID-19. As PA is important in maintaining health and deconditioning takes time to reverse, this decline may have long-term implications for PA and health.

## STRENGTHS AND LIMITATIONS OF THIS STUDY

⇒ A strength of this study was the comprehensive national cohort including all people registered for sick leave during the first pandemic wave in Sweden.

⇒ A limitation of this study was the less than 50% response rate to the 18-month follow-up survey, and differences between responders and non-responders need to be considered when interpreting the findings.

⇒ Another strength of this study was the use of a validated instrument to assess the physical activity (PA) level, with clearly defined questions that lend weight to the results and facilitate comparisons with other studies.

⇒ Another limitation was the retrospective assessment of PA level, asking responders to report their PA level 18 months previous, which involves risk or recall bias.

## INTRODUCTION

The COVID-19 pandemic has highlighted the importance of physical activity (PA) in maintaining physical and mental well-being while enduring lockdowns and periods of social isolation.[1 2] COVID-19 has been found to cause different acute symptoms[3] of varying severity; the vast majority of patients have a mild infection, but some have a more severe infection requiring acute hospitalisation.[4] For some, the symptoms from the acute COVID-19 infection remain, with a multitude of long-term consequences.[5 6] Limitations in usual activities have been reported as the third most prevalent symptom (14.9%) up to 1 year after infection in populations with both hospitalised and non-hospitalised individuals.[7] In a small Swedish study of individuals with a post-COVID-19 condition, 62% reported having difficulties being physically active 2 years postdischarge from hospital, though there was a significant improvement from the 4-month follow-up.[8]

Being regularly physically active is a well-known strategy recommended to promote health and prevent disease.[9] Regular PA can also strengthen the immune system, which has been suggested to be beneficial when exposed to viral communicable diseases,[10] including COVID-19.[11] When infected with COVID-19, physically inactive individuals are at higher risk of more severe illness,

hospitalisation and death than individuals who are regularly physically active.[11 12]

Several studies[13–18] have reported changes in PA during the pandemic. Although multiple studies in Swedish populations[13–15] have reported at least a slight increase in PA, other studies from the UK,[16] Italy[17] and USA[18] have reported a decrease. This may be partly explained by different approaches to restrictions during the pandemic.[14] The changes in PA also seem to depend on age,[13 16] sex,[13 17] occupation[13] and the different waves of the pandemic. A Swedish study[13] reported that more individuals changed their lifestyle habits during the first wave than during the second wave, with 26.4% reporting a negative change in daily activity during the first wave, compared with 20.3% during the second. Furthermore, studies have shown that decreased PA is associated with reduced life satisfaction[14] and a negative impact on mental health,[13 17 18] and that social inequality in PA increased during the pandemic.[15]

In the working-age population, falling ill with COVID-19 often means having to take sick leave from work.[19] For some, this results in long periods of sick leave and, sometimes, periods of recurrent sick leave.[20 21] In Sweden, compensation for sick leave is comprehensive, tax funded and administered by the Swedish Social Insurance Agency (SSIA). Our previous study found that nearly 12 000 individuals in Sweden were on sick leave due to COVID-19 within the framework of sickness benefits during the first pandemic wave.[20] These individuals comprise the cohort of the current study.

Based on the national cohort of individuals registered for sick leave due to COVID-19 during the first pandemic wave, we investigated how PA and changes in PA levels in regard to acute disease severity relate to perceived difficulties in performing daily life activities 18 months after a COVID-19 infection.

## MATERIALS AND METHODS
### Study population and setting
This study is a long-term follow-up survey of a national cohort that comprised all individuals in Sweden who took sick leave due to COVID-19 (11 955 individuals) during the first wave of the pandemic.[20] The inclusion criterion was sick leave due to a COVID-19 diagnosis (International Statistical Classification of Diseases code U07) registered in the national SSIA from 1 March through 31 August 2020. Registration with the SSIA starts at 2 weeks of absence due to sickness for individuals who are employed. If self-employed or unemployed, individuals are registered with the SSIA from the start of the absence. In this study, receiving sickness benefits was defined as sick leave regardless of the amount. The study combines long-term follow-up survey data and data from national registers.

### Patient and public involvement
A patient partner was not involved in the present study.

### Registry data
The SSIA provided the study population and sick-leave data for the study cohort. The sick-leave period due to COVID-19 was the number of days registered for the sickness benefits that included at least one registration with a COVID-19 diagnosis. Sick leave during the year prior to COVID-19 was defined as being registered for sick leave ≥28 days or ≥6 times between 1 March 2019, and the start of sick leave due to COVID-19. The National Board of Health and Welfare maintains the National Patient Registry of all inpatient care in Sweden. The Registry contributed data on in-hospital care due to COVID-19, which was defined as at least 1 day of hospital stay with registration of any U07 diagnoses. In the present study, in-hospital care due to COVID-19 was used as a proxy for acute disease severity; having been hospitalised defined more severe acute disease than not having been hospitalised. Statistics Sweden provided data on education and income levels; the individual disposable income during the year prior to COVID-19 (2019) was retrieved and presented in tertiles of low, medium and high income.

### Survey data
Approximately 18 months after the start of initial sick leave registration due to COVID-19, the follow-up survey was distributed to the cohort by the survey company Indicator (Institutet för kvalitetsindikatorer). The company only had access to postal addresses and telephone numbers for the cohort. A postal invitation letter to participate in the survey was first distributed on 25 February 2022. The options provided for completing the survey were over the web or by post. This was followed up with two reminders by text messaging and one by postal mail. The survey was open for completion until 24 April 2022. Informed consent was given by responding to the questionnaire. Survey data were merged with cohort data by code, and researchers only had access to pseudonymised data. The code key is kept by Statistics Sweden.

The questionnaire included several questions regarding remaining symptoms due to COVID-19, perceived recovery and functioning in daily life, and PA levels before COVID-19 and at follow-up. For the purpose of this study, questions on past and current PA levels, as well as an item on long-term consequences of COVID-19 on everyday life, were included.

Current PA level and PA level prior to COVID-19 were reported using the standardised Saltin-Grimby PA Level Scale (SGPALS),[22 23] which is scored at four levels: (1) physically inactive; (2) light PA for at least 4 hours/week; (3) regular moderate PA and training for at least 2–3 hours/week; and (4) regular vigorous physical training for competitive sports several times each week. The long-term consequences of COVID-19 were reported using the question, 'Do you have difficulties performing daily life activities due to COVID-19?' which was answered yes/no and/or by grading difficulties on a 3-point scale (small, some, or big difficulties). The variable was

dichotomised into having difficulties in daily life activities and not having difficulties in daily life activities.

## Statistical analysis

Data management and analyses were performed in SPSS Statistics V.25 (IBM). Descriptive statistics are presented as frequencies and percentages, means with SDs, or medians with IQR. Analyses of group comparisons, including a drop-out analysis, were conducted using the Student's t-test with continuous variables, the Mann-Whitney U test with ordinal variables, or $\chi^2$ test with categorical variables. Analyses of changes in PA within groups were performed using the Wilcoxon signed-rank test. A Sankey diagram was created to graphically illustrate changes in PA. Multiple binary logistic regression was performed to assess the association of changes in PA with perceived difficulties performing daily life activities, and the results are presented in a forest plot. Variables included in the model were change in PA level from prior COVID-19 infection up until the 18-month follow-up, age, sex, initial hospitalisation and sick leave during the year prior to COVID-19 infection. The significance level was set to $p < 0.05$.

## RESULTS

### Study sample

Out of 11955 individuals in the cohort, 5464 individuals completed the survey, corresponding to a response rate of 45.7%, 4240 of whom completed the survey via the web. The mean age of the responders was 51 years, 66% of whom were women. Among responders, the mean number of days on sick leave during the first year following COVID-19 infection was 67 (SD 91.8), and approximately one in four had been hospitalised due to COVID-19 (table 1).

Responders were older, had longer duration of sick leave due to COVID-19, were more likely to have been hospitalised due to COVID-19, were more often women, and had higher education and income levels than non-responders. There was no significant difference in sick leave prior to COVID-19 between responders and non-responders (table 1).

### Physical activity

PA level was lower in hospitalised responders than in non-hospitalised responder both prior to COVID-19 (p=0.035) and at the 18-month follow-up (p=0.008; table 2).

In the whole group of responders, the PA level decreased from prior COVID-19 infection to the 18-month follow-up (p<0.001; figure 1). Summing up the changes in PA, the level of PA decreased in 1726 (33.6%) responders, was sustained in 3105 (60.5%) responders and increased in 300 (5.8%) responders. Of the individuals with the highest level of PA (SGPALS 4) prior to COVID-19, approximately one-third sustained their

**Table 1** Characteristics of responders and non-responders

| | Responders n=5464 (45.7%) | Non-responders n=6491 (54.3%) | P value |
|---|---|---|---|
| Age, mean (SD), years | 50.85 (10.21) | 45.60 (11.62) | <0.001 |
| Sex, n (%) | | | <0.001 |
| Men | 1865 (34.1) | 2961 (45.6) | |
| Women | 3599 (65.9) | 3530 (54.4) | |
| Education level, n (%) | | | <0.001 |
| Primary school (≤9 years) | 347 (6.4) | 890 (13.9) | |
| Secondary school (10–12 years) | 2598 (47.6) | 3291 (51.3) | |
| Short university education (13–14 years) | 841 (15.4) | 902 (14.1) | |
| Long university education (≥15 years) | 1667 (30.6) | 1328 (20.7) | |
| Employment status, n (%) | | | <0.001 |
| Employed | 5384 (98.5) | 6306 (97.1) | |
| Unemployed | 80 (1.5) | 185 (2.9) | |
| Income, n (%) | | | <0.001 |
| Low income | 1428 (26.1) | 2555 (39.4) | |
| Medium income | 1900 (34.8) | 2085 (32.1) | |
| High income | 2136 (39.1) | 1849 (28.5) | |
| Hospitalisation due to COVID-19, n (%) | 1503 (27.5) | 1457 (22.4) | <0.001 |
| Days of sick leave | | | |
| Mean (SD) | 67.28 (91.80) | 57.80 (74.55) | <0.001 |
| Median (IQR) | 36 (28) | 35 (24) | 0.003 |
| Sick leave prior to COVID-19, n (%) | 856 (15.7) | 1076 (16.6) | 0.178 |

**Table 2** Current and previous physical activity levels in the responders

| | Responders | Responders hospitalised due to COVID-19 | Responders non-hospitalised due to COVID-19 | P value* |
|---|---|---|---|---|
| Physical activity level prior to COVID-19 infection | 5175 (94.7) | 1438 (95.6) | 3737 (94.3) | |
| Vigorous physical activity | 455 (8.8) | 111 (7.7) | 344 (9.2) | 0.035 |
| Moderate physical activity | 2085 (40.3) | 560 (38.9) | 1525 (40.8) | |
| Light physical activity | 2252 (43.5) | 642 (44.6) | 1610 (43.1) | |
| Physically inactive | 383 (7.4) | 125 (8.7) | 258 (6.9) | |
| Current physical activity level | 5187 (94.9) | 1437 (95.6) | 3750 (94.7) | |
| Vigorous physical activity | 159 (3.1) | 39 (2.7) | 120 (3.2) | 0.008 |
| Moderate physical activity | 1369 (26.4) | 395 (27.5) | 974 (26.0) | |
| Light physical activity | 2782 (53.6) | 726 (50.5) | 2056 (54.8) | |
| Physically inactive | 877 (16.9) | 277 (19.3) | 600 (16.0) | |

Values are given as n (%).
*Mann Whitney-U test.

level of PA until the 18-month follow-up. In the subgroup analysis based on acute disease severity, the PA level decreased in the non-hospitalised group (p<0.001), with 1244 responders (33.5%) decreasing their level of PA, 2282 (61.5%) sustaining their level of PA, and 184 (5.1%) increasing their level of PA. A significant decrease in PA was also found in the initially hospitalised responders (p<0.001), with 482 (33.9%) decreasing their level of PA, 823 (58.0%) sustaining their level of PA and 115 (8.1%) increasing their level of PA. There was no difference in change in PA over time between hospitalised and non-hospitalised responders (p=0.193).

### Long-term consequences of COVID-19 in daily life and relationship to PA

At the 18-month follow-up, 2434 responders (46%) reported still having difficulties performing daily life activities due to COVID-19. Logistic regression was performed to explain these long-term difficulties. The model showed that a decrease in PA increased the odds (OR 5.58, 95% CI 4.90 to 6.34) of having difficulties performing daily life activities. Furthermore, having been initially hospitalised due to COVID-19 and taking sick leave during the year prior to COVID-19 also increased the odds, whereas age and sex did not contribute significantly to long-term difficulties in daily life activities (figure 2). The model could significantly (p<0.001) distinguish between responders who did or did not report perceived difficulties in performing daily life activities at the 18-month follow-up.

### DISCUSSION

The results of this study show that individuals with less severe acute disease had a higher level of PA, both prior

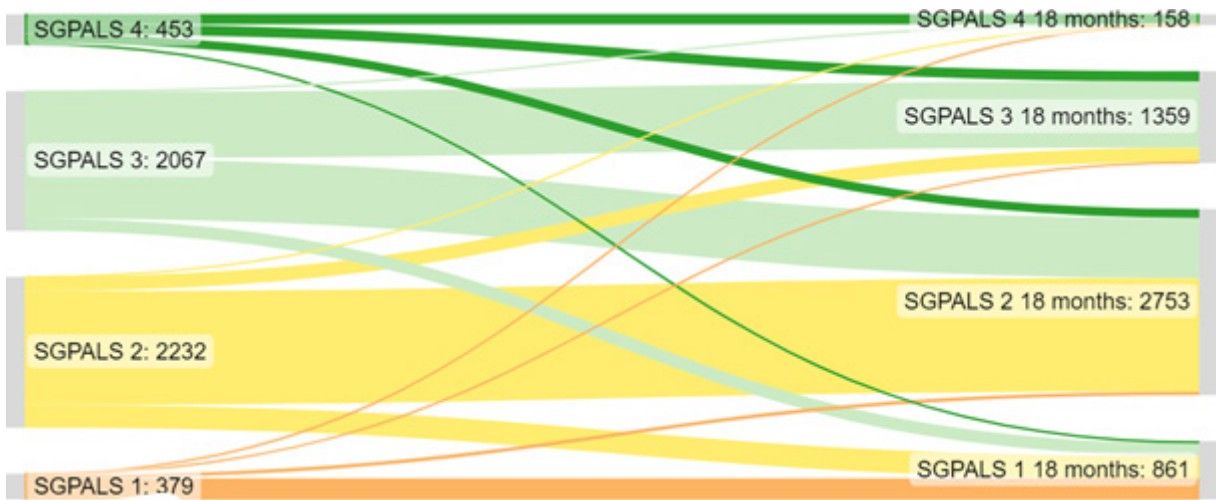

**Figure 1** Changes in physical activity (PA) levels according to the Saltin-Grimby PA Level Scale (SGPALS) from prior to COVID-19 infection (left side) to the 18-month follow-up (right side) n=5131 individuals. SGPALS: 1, physically inactive; 2, light PA for at least 4 hours/week; 3, regular moderate PA and training for at least 2–3 hours/week; and 4, regular vigorous physical training for competitive sports several times per week.

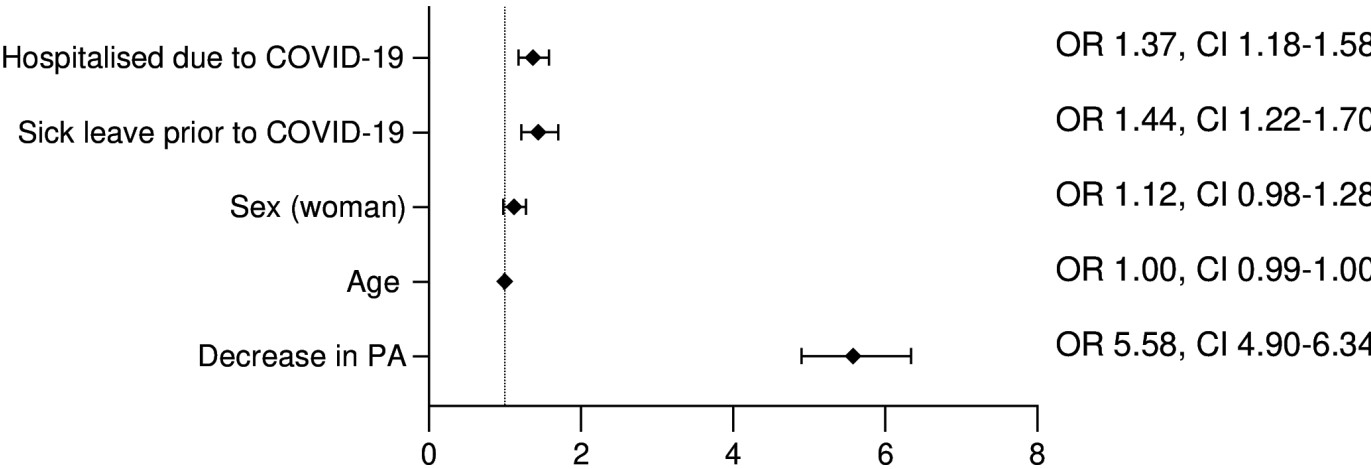

**Figure 2** Variables in the model explaining the odds of perceiving difficulties in performing daily life activities. The model explained between 15% (Cox & Snell $R^2$) and 20% (Nagelkerke $R^2$) of the variance in the outcome.

to COVID-19 and at the 18-month follow-up, than individuals with more severe acute disease. In both the total study population and both groups based on acute disease severity, the PA level was decreased at the 18-month follow-up compared with the level prior to COVID-19. Furthermore, a decrease in PA over that time was associated with perceived difficulties performing daily life activities 18 months after COVID-19.

The PA level of responders was rather high before COVID-19. The study population also seems to be quite physically active compared with other cohorts in which PA has been assessed by the same measure.[24–26] Given that regular PA has been shown to be beneficial in protecting individuals against severe illness and hospitalisation for COVID-19,[11 12] we could anticipate that this population would not be severely affected by COVID-19 infection. Nevertheless, one-fourth of the current study population was initially hospitalised. Being more physically active during the pandemic may have entailed being in places or contexts with a higher risk of transmission of COVID-19, such as gyms or team sports, whereas less active individuals may have been more isolated in their own homes. Furthermore, exercising while already infected with COVID-19 may have worsened the course of the disease.[27] In the polio epidemics, being very physically active[28] or pregnant,[29] both of which affect the immune system, increased the risk of paralysis among poliomyelitis cases. However, in the current study population's subgroups based on acute disease severity, hospitalised responders had significantly lower PA levels than non-hospitalised responders, supporting previous findings of the benefits of PA in regard to COVID-19 severity.[11 12]

Summing up the changes in PA, there was a decrease in the level of PA in both the total study population and the groups based on acute disease severity. In the general Swedish population, PA increased during the pandemic,[13–15] in contrast to the current findings in a cohort of individuals who had COVID-19. The decrease in PA seen in the current study can be speculated to relate to the disease itself and deconditioning.[30] However, a

Swedish study[14] reported that, in a general population, 36% reported increased PA and 30% decreased PA during the pandemic; 71% and 89%, respectively, attributed this change to COVID-19-related restrictions.[14] The relatively mild approach to restrictions in Sweden may have had diverse consequences for PA in different groups of individuals. Recommendations, such as working from home, may have given them more free time and, therefore, increased PA for some, whereas cancelled sporting events and competitions may have had the opposite effect, especially among individuals with a high level of PA. In the current study, only one-third of individuals with the highest level of PA prior to COVID-19 sustained their PA level until the 18-month follow-up.

Furthermore, the strictest recommendations, which involved isolation and were applied to people >70 years of age and those infected, as a means to protect the most vulnerable and constrain transmission, most likely had an influence on the levels of PA. A study from the US showed an association between self-isolation and negative mental health that was not seen with social distancing.[18] Previous studies[13 17 18] also reported an association between negative mental health effects and a decrease in PA during the pandemic. Taken together, these findings suggest that pandemic-related restrictions and mental health could also play a role in the decrease in PA in the current cohort.

In this study, in-hospital care due to COVID-19 was used as a proxy for acute disease severity. As no one in Sweden was denied in-hospital care due to lack of capacity and healthcare in Sweden is tax funded, this proxy for acute disease severity seems reasonable and relies on the assessment of educated healthcare professionals. Furthermore, need for hospital care is included in the WHO Clinical Progression Scale,[31] which supports the interpretation of in-hospital care as a proxy for COVID-19 disease severity.

Using the SGPALS,[23] the current study showed a decrease in PA among individuals who were on sick leave due to COVID-19 during the first wave of the pandemic in Sweden. Notably, only one-third

of individuals with the highest level of PA prior to COVID-19 sustained their PA level until the 18-month follow-up. Because making up for deconditioning takes time,[32] this decline in PA may have long-term implications, and returning to pre-COVID-19 activities and levels may be difficult even for the most highly trained individuals, such as elite athletes.[33]

PA is important for maintaining health and can protect against both non-communicable chronic diseases[34] and communicable diseases,[35] such as COVID-19. Therefore, the results of this study, with a decreased PA level regardless of acute disease severity, are worrisome, especially considering the risk of future pandemics.

A strength of the present study was that a validated instrument, the SGPALS, was used for questions regarding PA. Furthermore, the different steps of the scale were clearly defined in the questionnaire, making it easier for individuals to answer according to their true activity levels. This lends weight to the results and facilitates comparisons with other studies and populations. However, the retrospective assessment of PA level, asking responders to report their PA level 18 months previous, involves risk or recall bias, which is a limitation.

Another limitation of the study was the response rate of nearly 50%, and responders were found to be older and having had more severe acute disease than non-responders. Thus, differences between responders and non-responders need to be considered when interpreting the findings.

In conclusion, PA levels were reduced 18 months after COVID-19 infection. A decrease in PA over that time was associated with perceived difficulties performing daily life activities 18 months after COVID-19. As PA is important in maintaining health and deconditioning takes time to reverse, this decline may have long-term implications for PA and health.

**Contributors**  AP contributed to the conception and design of the study, managed data acquisition and analysis, contributed to interpreting the data and drafted the work. AP acts as guarantor for the study. JS contributed to interpreting and presenting the data, and drafting the work. HCP contributed to the conception, design and acquisition of the study, as well as data management and interpretation. KSS contributed to the conception, design and acquisition of the study, as well as interpretation of the data. All authors reviewed the work critically for important intellectual content and approved the final version to be published.

**Funding**  This study has been funded by the AFA Insurance (grant number 200324) and the Swedish state under the agreement between the Swedish government and county council, the ALF agreement (grant numbers 965653 and 942914).

**Competing interests**  None declared.

**Patient and public involvement**  Patients and/or the public were not involved in the design, or conduct, or reporting, or dissemination plans of this research.

**Patient consent for publication**  Not applicable.

**Ethics approval**  This study was approved by the Swedish Ethical Review Authority on 24 June 2020 (2020-03046), with an amendment approved on 1 September 2021 (2021-03556). Participants gave informed consent to participate in the study before taking part.

**Provenance and peer review**  Not commissioned; externally peer reviewed.

**Data availability statement**  Data are available upon reasonable request. Complete data cannot be made publicly available for ethical and legal reasons according to Swedish regulations in the "Act concerning the Ethical Review of Research Involving Humans (2006:460)" and the Swedish Ethical Reviews Authority (http://www.epn.se). Permission to use data is only for what has been given ethical approval by the Swedish Ethical Review Authority. Upon reasonable request, data may be available from the Department of Clinical Neuroscience, SU/Sahlgrenska, Blå stråket 7, plan 3, SE-413 45 Göteborg, Sweden. Email: inf@neuro.gu.se.

**ORCID iDs**
Annie Palstam http://orcid.org/0000-0002-7127-213X
Katharina S Sunnerhagen http://orcid.org/0000-0002-5940-4400

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
