## [Reviewer comments · BMJ Open]

ARTICLE DETAILS

TITLE (PROVISIONAL)	Physical activity, acute severity, and long-term consequences of COVID-19: an 18-month follow-up survey based on a Swedish national cohort
AUTHORS	Palstam, Annie; Seljelid, Johanna; Persson, Hanna; Sunnerhagen, Katharina

VERSION 1 – REVIEW

REVIEWER	Hansson, Eva Lunds Universitet, Department of Health Sciences
REVIEW RETURNED	02-Oct-2023

GENERAL COMMENTS	Thank you for giving me the opportunity to read and review this manuscript. The topic is relevant for the readers of BMJ Open. I congratulate the authors on a very well written manuscript and a very well performed study. I have only a few minor comments. Specific comments: Title: Adequate Abstract: Well written Introduction: Page 5, line 5 and 13: blank space is missing before references. Aim: Clear Method: Page 6: patient partner: If the mentioning of a patient partner in other research in the group should be meaningful, the authors need to add information regarding if the patient partner has in any way discussed or shared knowledge within this specific study. Otherwise, the information is irrelevant. Results: Well described and well displayed. Discussion: Page 10, line 36-37: blank space is missing before references. Page 10, line 56: the word “elderly” is considered ageism and should be avoided. Conclusion: Adequate and well written References: Adequate Tables and figures: Table 2: Please add the statistical method for calculating p-values in a footnote. For clarity: add “prior to Covid-19” on the y-axel in figure 1.
---

REVIEWER	Eek, Frida Lunds Universitet, Health Sciences
REVIEW RETURNED	11-Oct-2023

GENERAL COMMENTS	Thank you for the opportunity to read and review this manuscript. Consequences after the Covid pandemic is an interesting topic, and physical activity is a valuable general aspect for better health. This study aimed to examine associations between infection aspects, physical activity and health related functions, which is an
---

interesting topic. The study is based on a large sample of people with sick leave due to Covid during the first wave of pandemic. The collected data seems valuable and interesting; however I have some concerns and comments regarding formulated aims, analyses and conclusions.

First of all, I don't think you should make any statements regarding the effect/impact of (changes in) physical activity on function- there is no support for causal statements based on this design and analysis. Isn't it very likely that a more serious illness (there is a range of severity not just based on hospitalised or not), including post-covid, has affected both physical activity and functional ability? You can make statements about associations and differences, but not effects and impact. This applies both in the title, purpose and conclusions. See further detailed comments below:

The formulated purpose of the study gives the impression of carrying out a combined analysis of all three aspects (disease severity, changes in PA, and function), but that is not what you performed? You analysed (1) change in reported PA over time, with no analytical comparisons between severity groups, and (2) relation between change in PA and perceived function. I am aware that general aims are considered not to be adjusted "post hoc", but here it is more regarding clarification, and I would also suggest a better match between aim and analyses.

Abstract:

Abstract objective:

In abstract, the formulated aim is to investigate how changes in levels of physical activity (PA) relate to acute disease severity. However, the comparison is just made regarding the prior PA level and severity level (hospitalization), which is mentioned in title. The analysis of change is just performed in total and stratified within-group analyses, with no comparison of *change* between severe/hospitalized and non-hospitalized groups. Why did you not perform an analysis of difference in change (if there was an aim to examine how the severity was related to subsequent changes in PA)? It is however a bit unclear which aspect you consider as the exposure vs outcome.

One option could be to dichotomise the PA-change (e. g. as either decrease, or increase/no change) and perform a multivariable logistic regression with this variable as outcome. Or at least the "magnitude" of change could be compared between the groups. It would however be good to have the possibility to adjust for variables such as age (since I would guess it is likely that older people had higher disease severity, and it may also be more likely to decrease PA in higher ages) so a non-parametric comparison would not be optimal. However, again, it is not so clear to me which variable you consider as exposure vs outcome, and what the actual aim/research question is.

Abstract "Participants":

The formulation "The study combined register and survey data... WITH a follow up survey" sounds a bit unmatched with the cross-sectional design. Could you re-formulate the description? Or at least state that the study is based on the "follow up" survey. Maybe this description should be defined under design, and not "participants"? You also describe survey content under "participants". Isn't that more related to the outcomes? And under

	outcomes you describe statistical analyses.. Also, the 11955 are not the “participants”, since only 45% participated in the current study. I suggest that you re-consider the headline structure. The ”comparison of groups” is unclear -which groups are compared, regarding which outcomes? Does the logistic regression really confirm “The ‘value’ of change... ‘explaining’...”? Or does it more simply assess the relative odds or relation between defined independent and dependent variables? Abstract results: “A decrease in PA increased the odds...” As mentioned before, you should be careful to make causal/impact statements. It could just as well/likely be the more severe versions of covid (not just the hospitalized), or post covid, that causes both decrease in PA and the function. You can state that they had higher odds, but not that the decrease *affected* the odds/function. Abstract conclusion The conclusion should preferably be more clearly related/based on your results Strength and limitations You mention that differences between responders and non-responders needs to be considered when interpreting the findings. How/where did you consider the differences between responders and non-responders in the interpretation of results? (in discussion) P 6 r 29-32. Aim: As previously mentioned, the stated aim does not clearly relate to the analyses you perform. Isn't what you analyse: 1. Differences in reported PA level before and after covid between hospitalised and non-hospitalised individuals. 2. The (retrospective report of) change in PA, stratified by hospitalised and non hospitalised individuals. And 3. The relation between change in PA and current perceived daily function? If you have other aims, completing analyses should be performed. All performed and presented analyses and results should be clearly related to defined aim and/or research questions. Study population and design “The study is a long term follow up...” this doesn't either relate to cross sectional design. This cross sectional study is based on a follow up survey? P 7, r 19. The ethical approval is usually presented either initially in the methods section, or as a separate section at the end of the method section. Not stated in the middle of the survey description. You describe “survey data”, but I suggest that you clearly define the definitions of outcome and independent variables. E g define the “severity” variable, and also the binary definition you use for the daily function outcome (since you describe two different questions). Statistics
--	---

	I suggest you describe the analyses more clearly related to dependent and independent variables. E g “group comparisons” are unclear, since it’s not stated which groups are compared, regarding which outcomes. I assume that Mann-Whitney is used for at least/only the ordinal scales, not nominal, so that could be defined. As previously mentioned, -why did you not compare the change in PA? Regarding the multivariable logistic regression -why did you not include any other variable relating to the severity level, e g sick leave during the year following the infection rather than before the infection? (Or is the sick leave before the infection considered as a general health level? Then it’s also motivated to include.) But again, the severity of infection could vary more than just by hospitalization or not. Post-covid could e g result in both decreased PA and lower daily function? The perceived recovery (that was included in the survey) was not considered to include in analyses? Results: The participating group is the study sample rather than the population? (I’m aware that study population is often used, but since it is a sample and not a population, I prefer the term study sample) Add sd when you present mean values in text. Regarding the comparison of PA levels between hospitalised and non-hospitalised individuals -did you check if there are differences in age? Older people had higher risk for more severe infection, and may likely also have lower levels of PA. Page 10, r 36-37. Same comment as in abstract -avoid “effects” or causal interpretations. You mention “final” model -was it performed step wise? Describe in methods section. Discussion: I suggest you put the method discussion either in the beginning (following overall summary) or the end, not in the middle of the results discussion. But limitations of the study is not lifted? Conclusion: “The covid pandemic has spotlighted the importance of PA in maintaining physical and mental well being” -what is this conclusion based on? You don’t examine mental (or either physical) “well being”?
--	--

VERSION 1 – AUTHOR RESPONSE

Reviewer: 1

Dr. Eva Hansson, Lunds Universitet

Comments to the Author:	
Thank you for giving me the opportunity to read and review this manuscript. The topic is relevant for the readers of BMJ Open. I congratulate the authors on a very well written manuscript and a very well performed study. I have only a few minor comments.	We would like to thank reviewer 1 for positive feedback and valuable input.
Title: Adequate	OK
Abstract: Well written	Thank you
Introduction: Page 5, line 5 and 13: blank space is missing before references.	Thank you for noticing this, this has now been corrected.
Aim: Clear	OK
Method: Page 6: patient partner: If the mentioning of a patient partner in other research in the group should be meaningful, the authors need to add information regarding if the patient partner has in any way discussed or shared knowledge within this specific study. Otherwise, the information is irrelevant.	Thank you for this comment. The sentence has now been removed.
Results: Well described and well displayed.	Thank you
Discussion: Page 10, line 36-37: blank space is missing before references. Page 10, line 56: the word "elderly" is considered ageism and should be avoided.	Thank you for this comment. The sentence has now been changed accordingly: Furthermore, the strictest recommendations, which involved isolation and were applied to people >70 years of age and those infected
Conclusion: Adequate and well written	Thank you
References: Adequate	OK
Tables and figures: Table 2: Please add the statistical method for calculating p-values in a footnote.	A footnote has been added to Table 2: *Mann Whitney-U test
For clarity: add "prior to Covid-19" on the y-axel in figure 1.	Thank you for this important input. We agree with the need for clarification, however the length of "prior to COVID-19" would interfere with the visibility of the Sankey diagram if inserted within the figure. We have tried different solutions and come

	to the decision to clarify the figure legend as follows: FIGURE 1 Changes in physical activity (PA) levels according to the Saltin–Grimby PA Level Scale (SGPALS) from prior to COVID-19 infection (left side) to the 18-month follow-up (right side) $n = 5131$ individuals. SGPALS: 1, physically inactive; 2, light PA for at least 4 hours/week; 3, regular moderate PA and training for at least 2–3 hours/week; and 4, regular vigorous physical training for competitive sports several times per week.
--	--

Reviewer: 2
Dr. Frida Eek, Lunds Universitet

Comments to the Author: Thank you for the opportunity to read and review this manuscript. Consequences after the Covid pandemic is an interesting topic, and physical activity is a valuable general aspect for better health. This study aimed to examine associations between infection aspects, physical activity and health related functions, which is an interesting topic. The study is based on a large sample of people with sick leave due to Covid during the first wave of pandemic. The collected data seems valuable and interesting; however I have some concerns and comments regarding formulated aims, analyses and conclusions.	We would like to than reviewer 2 for valuable comments. Please find response to each comment respectively below. All changes made in the manuscript are marked with track changes.
First of all, I don't think you should make any statements regarding the effect/impact of (changes in) physical activity on function- there is no support for causal statements based on this design and analysis. Isn't it very likely that a more serious illness (there is a range of severity not just based on hospitalised or not), including post-covid, has affected both physical activity and functional ability? You can make	We agree with your comment and have changed the wording accordingly, consequently exchanging words like "impact" and "influence" to "associate" and "relate", throughout the manuscript.

statements about associations and differences, but not effects and impact. This applies both in the title, purpose and conclusions. See further detailed comments below:	
The formulated purpose of the study gives the impression of carrying out a combined analysis of all three aspects (disease severity, changes in PA, and function), but that is not what you performed? You analysed (1) change in reported PA over time, with no analytical comparisons between severity groups, and (2) relation between change in PA and perceived function. I am aware that general aims are considered not to be adjusted “post hoc”, but here it is more regarding clarification, and I would also suggest a better match between aim and analyses.	Thank you for pointing this out. The objective in the abstract has now been changed to be coherent with the aim in the main text, and for clarity reasons, as follows: To investigate how changes in levels of physical activity (PA) in regard to acute disease severity relate to perceived difficulties in performing daily life activities 18 months after COVID-19 infection.
Abstract: Abstract objective: In abstract, the formulated aim is to investigate how changes in levels of physical activity (PA) relate to acute disease severity. However, the comparison is just made regarding the prior PA level and severity level (hospitalization), which is mentioned in title. The analysis of change is just performed in total and stratified within-group analyses, with no comparison of *change* between severe/hospitalized and non-hospitalized groups. Why did you not perform an analysis of difference in change (if there was an aim to examine how the severity was related to subsequent changes in PA)? It is however a bit unclear which aspect you consider as the exposure vs outcome. One option could be to dichotomise the PA-change (e. g. as either decrease, or increase/no change) and perform a multivariable logistic regression with this variable as outcome. Or at least the “magnitude” of change could be compared between the groups. It would however be good to have the possibility to adjust for	Thank you for this comment, and for interesting suggestions on perspectives on how to further explore PA. For clarification, the title has now been changed: “Physical activity, acute severity, and long-term consequences of COVID-19: an 18-month follow-up survey based on a Swedish national cohort” Further, the objective in the abstract was clarified and is now coherent with the aim in the main text, as stated in response to a previous comment. Since PA has been found to be of importance for disease severity, as stated in the introduction, we wanted to analyse if there was a difference in level of PA prior to COVID-19 between people who had been hospitalized and people who had not been hospitalized (proxy for disease severity). We were also interested to see how PA changed over time in the population with COVID-19, since the literature shows divergent results on this and a large cohort like this could contribute with knowledge on this. It was not important for us to analyze differences between hospitalized and non-

variables such as age (since I would guess it is likely that older people had higher disease severity, and it may also be more likely to decrease PA in higher ages) so a non-parametric comparison would not be optimal. However, again, it is not so clear to me which variable you consider as exposure vs outcome, and what the actual aim/research question is.	hospitalized people but rather to compare before and after in each group and illustrate the change over time. Since a large group of people with COVID-19 have trouble in their daily life activities years after their acute illness, we also wanted to investigate if daily life activities in the long-term could be associated with change in physical activity over time, and of course also include other variables of anticipated importance in the model, such as age, disease severity and others.
Abstract "Participants": The formulation "The study combined register and survey data... WITH a follow up survey" sounds a bit unmatched with the cross-sectional design. Could you re-formulate the description? Or at least state that the study is based on the "follow up" survey. Maybe this description should be defined under design, and not "participants"? You also describe survey content under "participants". Isn't that more related to the outcomes? And under outcomes you describe statistical analyses.. Also, the 11955 are not the "participants", since only 45% participated in the current study. I suggest that you re-consider the headline structure.	Thank you for pointing this out and for valuable suggestions. The Design section in the abstract has now been adjusted as follows: An observational study with an 18-month follow-up survey based on registry data from a national cohort. Further, the Participants section in the abstract has been changed as follows: "5464 responders to the 18-month follow-up survey of a Swedish national cohort of 11,955 individuals on sick leave due to COVID-19 during the first wave of the pandemic." The last sentence in the Participants section was moved to the Outcomes section: "The follow-up survey included questions on daily life activities, as well as present and retrospective level of PA." The headlines in the abstract are provided by the journal, why there was no separate headline for statistical analysis and we thought it was the best fit in the Outcomes section.
The "comparison of groups" is unclear - which groups are compared, regarding which outcomes?	With regards to the word limits in the abstract, this is instead further elaborated on in the methods section in the main text.
Does the logistic regression really confirm "The 'value' of change... 'explaining'..."? Or does it more simply assess the relative	Thank you for this comment, the wording has been changed as follows: Multiple binary logistic regression was performed to assess the association of changes in PA

odds or relation between defined independent and dependent variables?	with perceived difficulties in performing daily life activities.
Abstract results: “A decrease in PA increased the odds...” As mentioned before, you should be careful to make causal/impact statements. It could just as well/likely be the more severe versions of covid (not just the hospitalized), or post covid, that causes both decrease in PA and the function. You can state that they had higher odds, but not that the decrease *affected* the odds/function.	Thank you for this comment. You make an important point here, and this is why we also included age, sex, hospitalization (as a proxy for disease severity), and previous history of sick leave (as a proxy for prior health state) in the model to adjust for such aspects. All the same, change in PA was the variable that stood out with a high contribution to explaining the functioning after 18 months.
Abstract conclusion The conclusion should preferably be more clearly related/based on your results	Thank you for this comment. The conclusion has been adjusted to be more clearly related to the results as follows: PA levels were reduced 18 months after COVID-19 infection. A decrease in PA over that time was associated with perceived difficulties performing daily life activities 18 months after COVID-19. As PA is important in maintaining health and deconditioning takes time to reverse, this decline may have long-term implications for PA and health.
Strength and limitations You mention that differences between responders and non-responders needs to be considered when interpreting the findings. How/where did you consider the differences between responders and non-responders in the interpretation of results? (in discussion)	Thank you for this suggestion. We have now elaborated on the methods discussion as follows: A strength of the present study was that a validated instrument, the SGPALS, was used for questions regarding PA. Furthermore, the different steps of the scale were clearly defined in the questionnaire, making it easier for individuals to answer according to their true activity levels. This lends weight to the results and facilitates comparisons with other studies and populations. However, the retrospective assessment of PA level, asking responders to report their PA level 18 months previous, involves risk or recall bias, which is a limitation. Another limitation of the study was the response rate of nearly 50%, and responders were found to be older and having had more severe acute disease than non-responders. Thus, differences between

	responders and non-responders need to be considered when interpreting the findings.
P 6 r 29-32. Aim: As previously mentioned, the stated aim does not clearly relate to the analyses you perform. Isn't what you analyse: 1. Differences in reported PA level before and after covid between hospitalised and non-hospitalised individuals. 2. The (retrospective report of) change in PA, stratified by hospitalised and non hospitalised individuals. And 3. The relation between change in PA and current perceived daily function?	Thank you for this comment. It is correct that these specific steps are what we aim to investigate. We strived for a more inclusive aim to cover these specific analyses in one sentence. For clarity, based on your suggestions, we have now added to the aim as follows: Based on the national cohort of individuals registered for sick leave due to COVID-19 during the first pandemic wave, we investigated how PA and changes in PA levels in regard to acute disease severity relate to perceived difficulties in performing daily life activities 18 months after a COVID-19 infection.
If you have other aims, completing analyses should be performed. All performed and presented analyses and results should be clearly related to defined aim and/or research questions.	We agree with this and have no further aims or analyses to present.
Study population and design "The study is a long term follow up..." this doesn't either relate to cross sectional design. This cross sectional study is based on a follow up survey?	Thank you for your previous comment on this, with regards to the abstract. The description of the study design was changed according to your previous comment in the abstract, and is now coherent with the description in the main text.
P 7, r 19. The ethical approval is usually presented either initially in the methods section, or as a separate section at the end of the method section. Not stated in the middle of the survey description.	Thank you for this suggestion. The ethical approval is now moved to directly under the heading Data collection as follows: The study was approved by the Swedish Ethical Review Authority on June 24, 2020 (2020-03046), with an amendment approved on September 1, 2021 (2021-03556).
You describe "survey data", but I suggest that you clearly define the definitions of outcome and independent variables. E g define the "severity" variable, and also the binary definition you use for the daily function outcome (since you describe two different questions).	Thank you for bringing this to our attention. The definition of the severity variable has now been adjusted as follows: The Registry contributed data on in-hospital care due to COVID-19, which was defined as at least 1 day of hospital stay with registration of any U07 diagnoses. In the present study, in-hospital care due to COVID-19 was used as a proxy for acute disease severity; having

	been hospitalised defined more severe acute disease than not having been hospitalised. Further, the dichotomization of the daily functioning outcome has been clarified as follows: The long-term consequences of COVID-19 were reported using the question, “Do you have difficulties performing daily life activities due to COVID-19?” which was answered yes/no and/or by grading difficulties on a 3-point scale (small, some, or big difficulties). The variable was dichotomised into having difficulties in daily life activities and not having difficulties in daily life activities.
Statistics I suggest you describe the analyses more clearly related to dependent and independent variables. E.g. “group comparisons” are unclear, since it’s not stated which groups are compared, regarding which outcomes. I assume that Mann-Whitney is used for at least/only the ordinal scales, not nominal, so that could be defined.	Thank you for bringing to our attention need of clarification regarding statistical analyses. It would be too long description to describe each group comparison by specific test given the many tests performed in the drop-out analysis. However, for clarity, we have now added a footnote under table 2, clarifying that the Mann Whitney-U test was performed to analyze differences in PA levels between groups depending on acute disease severity. Please find this added directly under table 2, as suggested by reviewer 1. We have also adjusted the text in Statistics according to your suggestion, thank you for noticing this mistake. Analyses of group comparisons, including a drop-out analysis, were conducted using the Student t-test with continuous variables, the Mann–Whitney U test with ordinal variables, or chi-squared test with categorical variables.
As previously mentioned, -why did you not compare the change in PA?	Thank you for this question. We have now performed the analysis: group comparison using Mann-Whitney U test between hospitalised and non-hospitalised responders on change in PA over time. There is no change in the statistics section since the wording covers this added

	analysis. In the results section, this sentence has been added: There was no difference in change in PA over time between hospitalised and non-hospitalised responders (p=0.193).
Regarding the multivariable logistic regression -why did you not include any other variable relating to the severity level, e.g. sick leave during the year following the infection rather than before the infection? (Or is the sick leave before the infection considered as a general health level? Then it's also motivated to include.) But again, the severity of infection could vary more than just by hospitalization or not. Post-covid could e.g. result in both decreased PA and lower daily function? The perceived recovery (that was included in the survey) was not considered to include in analyses?	Thank you for this comment. It is correct that sick leave before the infection was included in the model as it is considered a proxy for general health level. It is correct that severity could vary more than just by hospitalization. Hospitalization is included in the WHO classification of disease severity in COVID-19, but we are aware that there are more nuances to disease severity than hospitalization or not. The question on perceived recovery was a question that was very similar to the question of having difficulties in performing daily life activities 18 months after initial infection. Therefore it was never considered to be included in the model.
Results: The participating group is the study sample rather than the population? (I'm aware that study population is often used, but since it is a sample and not a population, I prefer the term study sample)	Thank you for this suggestion. We agree with you on this being more clear and have changed the heading according to this suggestion.
Add sd when you present mean values in text.	The SD has now been added as follows: "Among responders, the mean number of days on sick leave during the first year following COVID-19 infection was 67, (SD 91.8), and approximately one in four had been hospitalised due to COVID-19 (Table 1)."
Regarding the comparison of PA levels between hospitalised and non-hospitalised individuals -did you check if there are differences in age? Older people had higher risk for more severe infection, and may likely also have lower levels of PA.	Thank you for this comment. It is known that people of older age are usually less physically active, and also at higher risk of being hospitalized. . Since age is a factor of importance, age was included as an independent variable in the logistic regression model together with PA, to adjust for such relations. In the model, change in PA stood out in contributing to the association with the outcome variable of functioning in daily life. Unfortunately, we

	did not analyze differences in age between hospitalized and non-hospitalized individuals in this study.
Page 10, r 36-37. Same comment as in abstract -avoid “effects” or causal interpretations. You mention “final” model -was it performed step wise? Describe in methods section.	The wording has been changed: Furthermore, a decrease in PA over that time was associated with perceived difficulties performing daily life activities 18 months after COVID-19. Also: The relatively mild approach to restrictions in Sweden may have had diverse consequences for PA in different groups of individuals. Recommendations, such as working from home, may have given them more free time and, therefore, increased PA for some... Thank you for noticing the fault to include the wording “final” model. There was no stepwise analysis, why the word “final” has now been admitted from the results section.
Discussion: I suggest you put the method discussion either in the beginning (following overall summary) or the end, not in the middle of the results discussion. But limitations of the study is not lifted?	Thank you for this suggestion of structure. The methods discussion has now been moved to the bottom, just above conclusion and elaborated as follows: A strength of the present study was that a validated instrument, the SGPALS, was used for questions regarding PA. Furthermore, the different steps of the scale were clearly defined in the questionnaire, making it easier for individuals to answer according to their true activity levels. This lends weight to the results and facilitates comparisons with other studies and populations. However, the retrospective assessment of PA level, asking responders to report their PA level 18 months previous, involves risk or recall bias, which is a limitation. Another limitation of the study was the response rate of nearly 50%, and responders were found to be older and having had more severe acute disease than non-responders. Thus, differences between

	responders and non-responders need to be considered when interpreting the findings.
Conclusion: “The covid pandemic has spotlighted the importance of PA in maintaining physical and mental well being” -what is this conclusion based on? You don’t examine mental (or either physical) “well being”?	Thank you for this comment. That phrase has been omitted from the conclusion. The conclusion has been changed as follows: In conclusion, PA levels were reduced 18 months after COVID-19 infection. A decrease in PA over that time was associated with perceived difficulties performing daily life activities 18 months after COVID-19. As PA is important in maintaining health and deconditioning takes time to reverse, this decline may have long-term implications for PA and health.